# Optical and Thermal Investigations of Eutectic Metallomesogen Mixtures Based on Salicylaldiaminates Metal Complexes with a Large Nematic Stability Range

Hassan-Ali Hakemi [1,†], Valentina Roviello [2,†] and Ugo Caruso [3,*]

1   Plastic Liquid Crystal Technology, Via Lambro 80, 20846 Macherio (MB), Italy
2   Department of Chemical, Materials and Industrial Production Engineering (DICMaPI),
    University of Naples Federico II, 80125 Naples, Italy
3   Department of Chemical Sciences, University of Napoli Federico II, Strada Comunale Cinthia, 26,
    80126 Napoli, Italy
*   Correspondence: ugo.caruso@unina.it
†   These authors contributed equally to this work.

**Abstract:** The mesomorphic behavior and the miscibility properties of binary mixtures of a new series of Schiff base metallomesogen (MOM) are evaluated by differential scanning calorimetry (DSC) and polarized optical microscopy (POM). Nuclear magnetic resonance (NMR), elemental analysis (CHNX), Fourier-transform infrared spectroscopy (FTIR) and thermogravimetric analysis (TGA) were used to certify the molecular structure of the compounds. The results revealed that the studied mixtures are completely miscible throughout the composition field and exhibit a nematic phase which covered the whole composition range. In the mixtures, the stability of the nematic phase varies continuously, and it is possible to highlight the presence of a eutectic composition with a wide mesogenic stability range.

**Keywords:** binary phase diagram; metallomesogen (MOM); induced phase; eutectic composition; mesophase stability range

## 1. Introduction

The studies on liquid-crystalline organo-transition metal complexes continue to generate enormous research interest for their ability to easily tune photochemical and photophysical properties. This has made them particularly promising for applications in organic electronics and sensors in various fields as photovoltaic devices [1–4], DNA binders [5–11], sensors [12–15], photosensitizers [16–18], molecular chemistry [19], fluorescent and colorimetric probes [20–31], medicinal chemistry [32] and metal-organic framework (MOF) construction [33–35] but also in more interesting topics of contemporary applied research such as electroluminescent displays, smart sensors, encryption systems or fuel cells but also in large varieties of metallomesogens with photoluminescence [36–38], electroluminescence [39–42], magnetic [43–45] or electric properties [46–48].

The main requirements for materials in technological applications are essentially due to a strong coupling of the electronic system and an efficient manipulation of the molecular orientation, both satisfied by MOMs.

In the MOMs, the strong coupling affects the ligand-metal system and occurs between the d orbital of the transition metal and the delocalized electronic system of ligands. It is attributable to the Intramolecular Charge Transfer (ICT) and to the charge transfer from metal to ligand (MLCT) or from ligand to metal (LMCT). The molecular orientation and its easy tunability is one of the main characteristics of liquid-liquid-crystalline materials [49,50]. Furthermore, the large birefringence and the possibilities of the efficient changes of the director under the electric field make MOMs ideal candidates for electronic devices [51].

Despite these promising premises, to date, there is no concrete evidence for the use of "metallomesogens" (MOM) as commercial materials. This is attributable to the lack of sufficient knowledge about the property/structure relationships necessary for the development of suitable materials and to their rather high transition temperatures that place serious limitations on their use. A valid approach to overcome these difficulties, alternative and more advantageous to molecular engineering and synthetic strategies, is through mixtures of multicomponent mesogenic systems [52–54]. In fact, the almost linear variation of the isotropization temperature as the composition and the "eutectic" trend of the melting temperature increase the stability range of the mesophase, making it considerably wider for the eutectic mixture than for pure components.

The present study aims to prepare two binary mixtures of MOMs derived from Schiff bases, different in type and length of the side chains and/or in the metal, with the aim to help us understand this phenomenology and improve our knowledge for a potential use of MOMs in a wide range of electronic devices. In the following sections, we will describe our experimental strategy based on some biligand calamitic nematic MOM model systems and their binary phase diagrams to develop eutectic mixtures with a low melting temperature and wide nematic range. This study is one of the first reports relating to our extensive industrial research and development programs on the application of MOM materials.

## 2. Results and Discussion

Liquid-crystalline and particularly metallomesogenic compounds are known to exhibit efficient control of molecular orientation by electrical tuning. In this perspective and with the aim of making possible the anchoring of such liquid-crystalline mixtures to polymeric supports in the form of films as a possible photoactive layer for electronic devices, we have synthesized three new metallomesogens with general Scheme 1.

**12-8NM**　　　 with R=O(CH$_2$)$_{11}$CH$_3$; R'=(CH$_2$)$_7$CH$_3$ and M=Cu
**A6O-8NM**　　with R=O(CH$_2$)$_6$OOCCH=CH$_2$; R'=(CH$_2$)$_7$CH$_3$ and M=Cu
**A11O-6ONM** with R=O(CH$_2$)$_{11}$OOCCH=CH$_2$; R'=(CH$_2$)$_3$OCH$_2$CH$_3$ and M=Ni, Pd

**Scheme 1.** General formula of metallomesogen (MOM) compounds.

Those compounds pertain to the class of Schiff bases complexes. Two of them have an acrylic function in the alkyl chain that is stable to thermal cycles, as demonstrated by the repeatability of thermal behavior in subsequent heating and cooling cycles.

The complex's formation is clearly proved by FTIR spectra. In Figure 1 it is reported, as example, the FTIR pattern of 12-8N and of its copper complex. It is evident that the band at 3422 cm$^{-1}$ assigned to $\nu$(OH) (intramolecular hydrogen bonded) disappears in the spectra of the Cu complex that indicates the deprotonation of the Schiff base and the formation of a bond with metal ion. In addition, the peak at 1718 cm$^{-1}$ assigned to $\nu$(C=N) in the free ligand system goes through a red shift to 1726 cm$^{-1}$ [55,56]. The same behavior is showed by all prepared complexes (Figures S1 and S2).

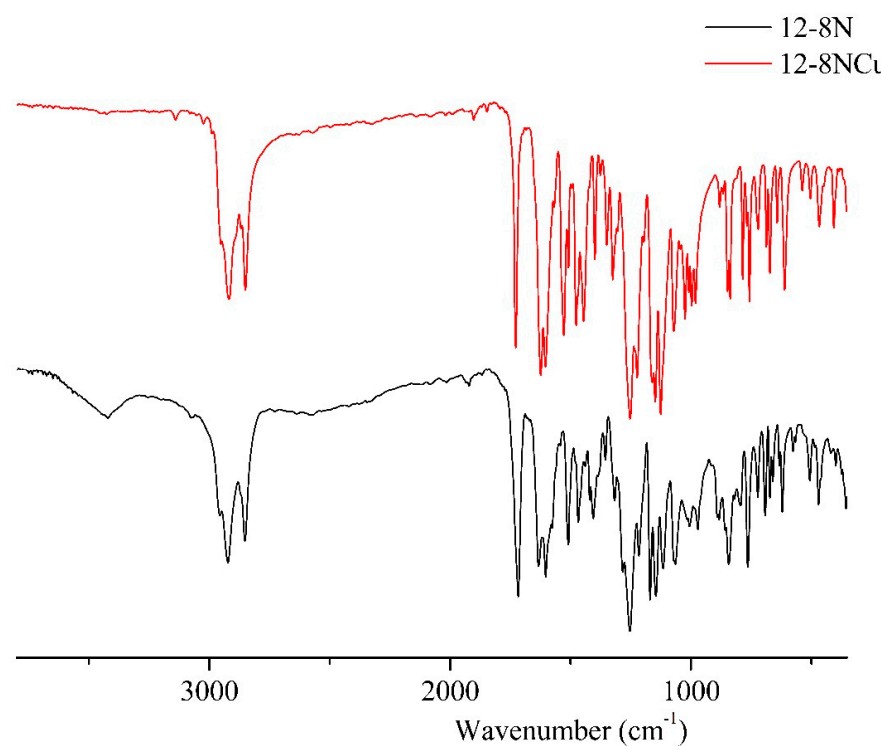

**Figure 1.** FTIR curves of 12-8N (black) and 12-8NCu (red).

The $^1$H-NMR of A11O-6ON and of its palladium complex are reported in Figure 2 as an example. The shift of the proton of imine, from 3.68 ppm in the free ligand to 4.11 ppm of the Pd compound is evidence of the complex formation. Analogue's behavior is observable in 1H NMR of A11O-6ONNi where the proton of imine shifts from 3.68 ppm in the free ligand to 4.03 ppm (Figure S3). In Table S1, the most relevant 1H NMR data are reported.

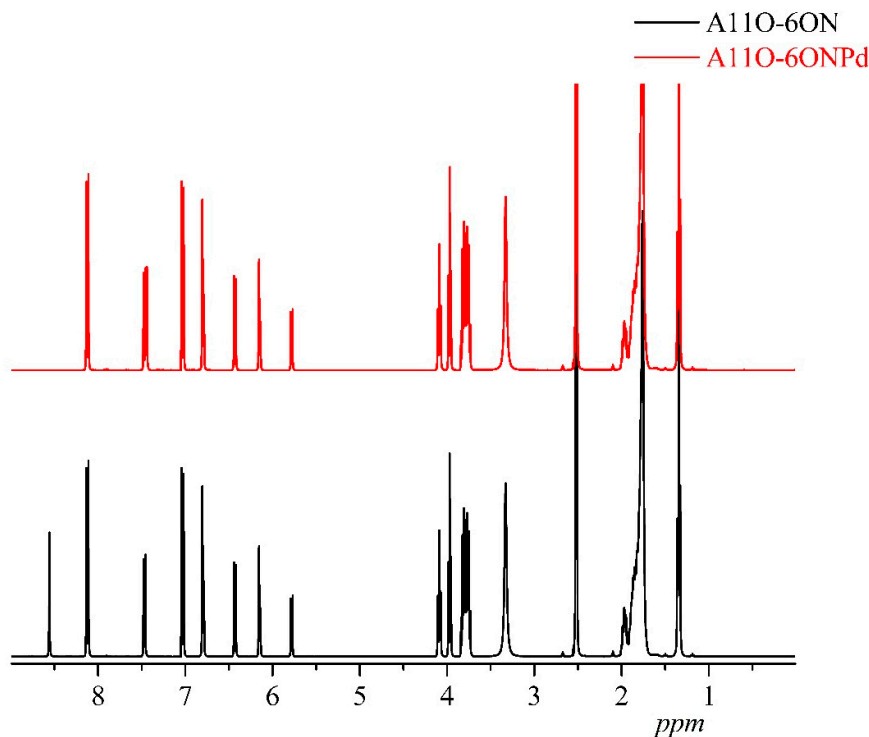

**Figure 2.** $^1$H NMR curves of A11O-6ON (red) and A11O-6ONPd (black).

Elemental analysis, metal content and NMR (Figures S4 and S5) are consistent with the assigned formula for all synthetized compounds.

As showed in Table 1, those that collect the transition temperature of synthetized MOM, all compounds showed a high stability range of the mesophase ranging between 80 and 150 °C. According to the choices made for the mixtures, which we will discuss briefly later, the temperatures shown in the table refer to the second heating cycle.

**Table 1.** Transition temperatures of MOMs.

| Compound | $T_m$ [1] | $T_i$ [2] |
|---|---|---|
| 12-8NCu | 103.5 | 129.0 |
| A6O-8NCu | 81.6 | 116.0 |
| A11O-6ONNi | 106.7 | 124.2 |
| A11O-6ONPd | 111.9 [c] | 143.6 |

[1] $T_m$ = melting temperature in °C; [2] $T_i$ = isotropization temperature in °C; [c] = solid−solid transition at 88.4 °C.

The liquid crystal behavior was easily detected by calorimetry, polarizing microscopy and X-ray diffraction techniques. Figure 3 shows a significant texture containing a nematic schlieren pattern of a 12-8NCu/A6O-8NCu binary mixture with a composition of 25% in 12-8NCu at 98.8 °C.

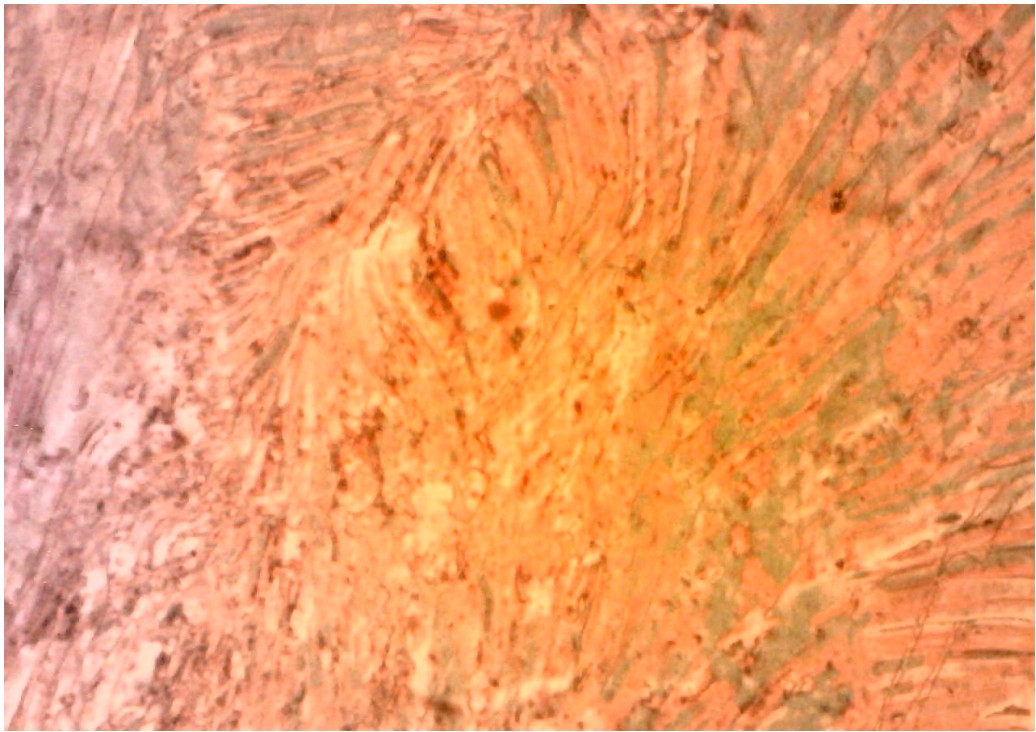

**Figure 3.** 12-8NCu/A6O-8NCu binary mixture with composition of 25% in 12-8NCu at 98.8 °C nematic schlieren texture. Crossed polarizers.

In Figures 4 and 5, there are reported examples of DSC thermograms for A11O-6ONNi and A11O-6ONPd. In the first heating run (black curve), the first endothermic signals correspond to the solid phase transition followed by recrystallization exotherms and from the melting peaks. The last endothermal signals attribute to the isotropization temperatures. At the cooling rates of 10 K/min, the thermograms (red curve) show two exothermic signals referable to anisotropization and recrystallization. The latter in the A11O-6ONNi thermogram is not completely reversible, as can be deduced from the presence of a shoulder on the melting peak in the second heating cycle (blue curve).

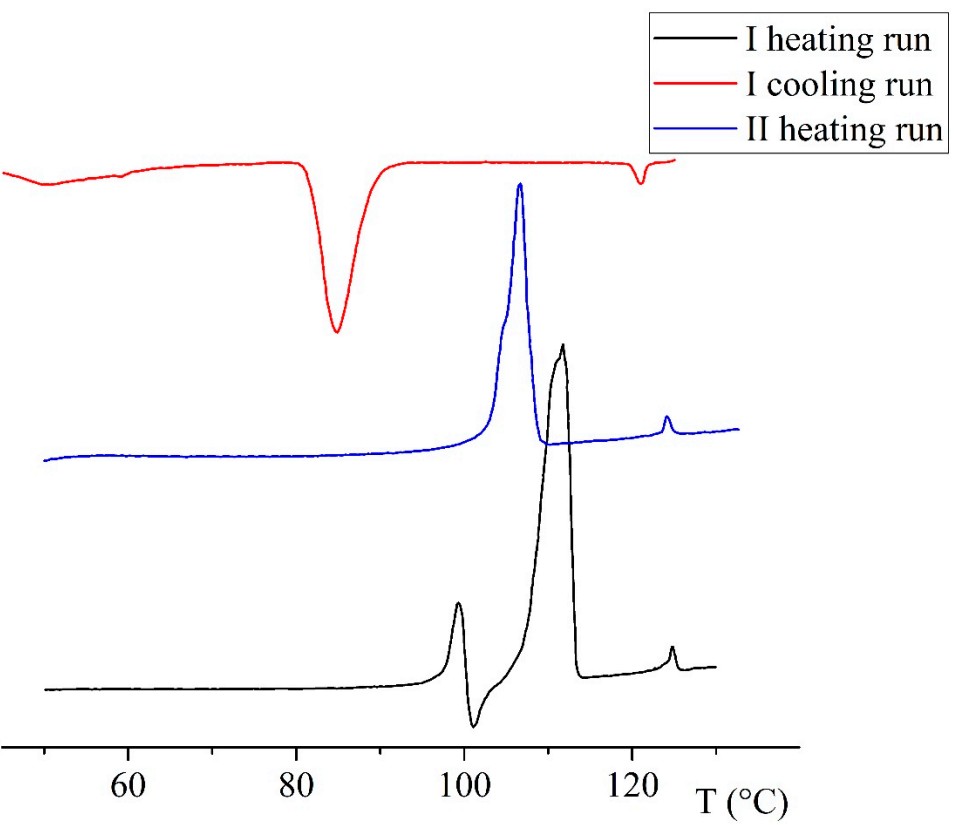

**Figure 4.** DSC thermograms of A11O-6ONNi: (black) first heating; (red) first cooling run; and (blue) second heating run at 10 K/min scanning rate.

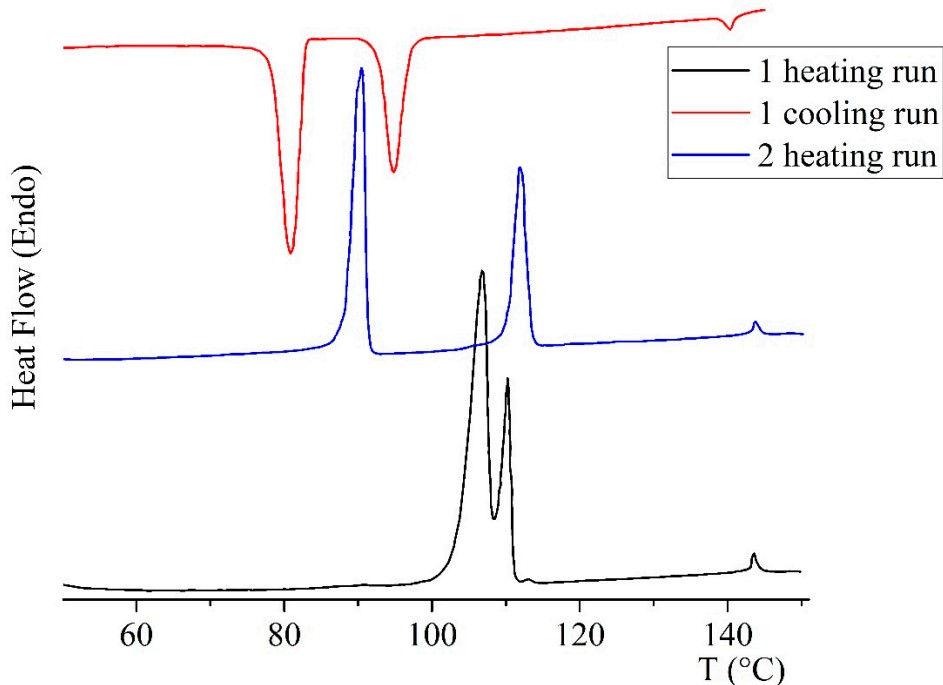

**Figure 5.** DSC thermograms of A11O-6ONPd: (black) first heating; (red) first cooling run; and (blue) second heating run at 10 K/min scanning rate.

The main differences are the presence of a solid-solid transition and transition temperatures exceed those of the corresponding nickel containing homologues. Analogous

differences were observed by us [57] and other scholars [58]. An explanation of this behavior could be linked to the greater stability of the Pd(II) complexes towards a pseudo-tetrahedral distortion of the planar square coordination geometry.

It is well known that the formulation of multi-component mixtures affects the transition temperatures, especially in the case of eutectic composition. In the case of liquid-crystalline mixtures' viscosity, permittivity, birefringence, etc., they also undergo a considerable variation. For this reason, we have prepared and studied two binary MOM mixtures.

The phase diagrams of the mixtures were obtained by differential scanning calorimetry (DSC) and were carried out by directly weighing the components in the DSC pans through repeated heating and cooling cycles with a scanning speed of 10 °C/min and 5 °C/min, respectively, considering that complete mixing took place when two successive thermal cycles at the same scanning rate provided the same thermal behavior.

To optimize the mixing of the components and ensure their uniform mixing, it was preferred to use the calorimetric data of the second heating cycle. By analogy, the same choice was made for the temperatures of the pure compounds (Table 1).

In the first blend system, components differ for type and length of lateral substituents linked to the same rigid core. All relevant thermal data are reported in Table 2 and Figure 6. The trend of the isotropization temperature, shown in Figure 4, increases regularly with the concentration of the 12-8NCu. Presumably, the lower isotropization temperature of A6O-8NCu is due to the greater flexibility and sinuosity of the side chain of the salicylic terminal.

**Table 2.** Transition temperatures of binary mixtures of 12-8NCu/A6O-8NCu in weight percent ratio.

| 12-8NCu (%wt.) | $T_m$ [1] | $T_i$ [2] |
|:---:|:---:|:---:|
| 0 | 81.6 | 116.0 |
| 25 | 71.2 | 117.7 |
| 50 | 80.8 | 122.0 |
| 75 | 90.8 | 124.4 |
| 100 | 103.5 | 129.0 |

[1] $T_m$ = crystal−nematic and [2] $T_i$ = nematic−isotropic transition temperatures in °C.

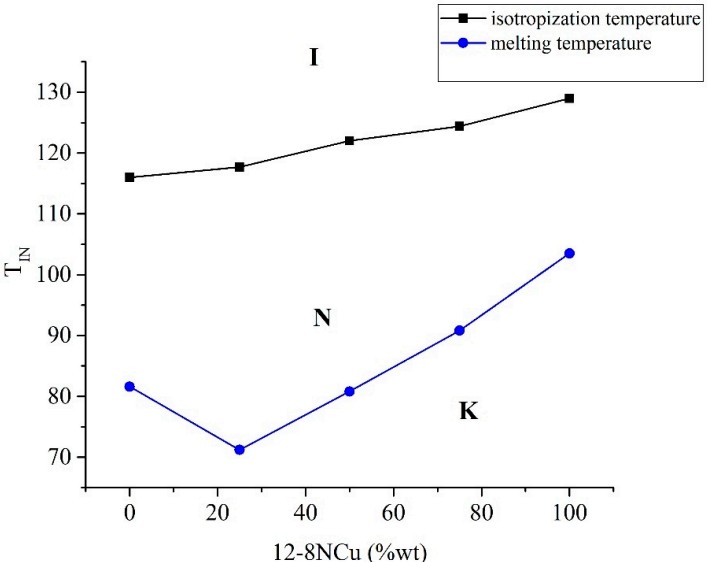

**Figure 6.** Binary phase diagram for 12-8NCu/A6O-8NCu blends. K indicates the stability field of the crystalline solid phase, N that of the nematic phase and I the isotropic phase.

In addition, the nematic phase is always observed showing the perfect miscibility of the two components throughout the composition field. The variation in crystallization temperature, as often observed, has a minimum for the wt./wt. composition of 0.25 to

a eutectic stability range of 46.5 °C, much wider than that of the two components 12-8NCu and A6O-8NCu of 25.5 °C and 34.4 °C, respectively. It is well known [59,60] that mesomorphic stability depends mainly on intermolecular interactions, in which the polarity of the molecule plays a predominant role and that the latter is strictly dependent on the nature and orientation of the lateral substituents [61,62].

In the second system, whose relevant thermal data are reported in Table 2 (Figures S6–S9), the binary phase diagram (Figure 5) was made from two metallomesogens which differ in the metal to which the salicylaldimine ligand is complexed.

Also, in this case, the stability of the nematic phase of both constituents is regularly affected upon mixing and decreases with wt.% of A11O-6ONNi (Table 3 and Figure 7). The eutectic mixture appears around a weight composition of 40% and shows a stability range of mesophase of 46.1 °C versus a stability range of 24.7 °C and 17.7 °C of A11O-6ONPd and A11O-6ONNi, respectively. As previously stated, the increased stability of the mesophase is correlated mainly by intermolecular interactions and then to polarity of the liquid-crystalline system, mainly determined to the orientation of lateral substituents. However, the steric factors that control packing, and then the intermolecular separation, play a dominant role in determining the type and stability of mesophase [63]; hence, it is the structural differences in the rigid core of the molecules that affect the mesomorphic properties of this system.

**Table 3.** Transition temperatures of A11O-6ONNi/A11O-6ONPd binary blends.

| A11O-6ONNi (%wt.) | $T_m$ [1] | $T_i$ [2] |
|---|---|---|
| 0 | 111.9 | 143.6 |
| 20.7 | 97.8 | 138.9 |
| 40.1 | 88.7 | 135.2 |
| 60.6 | 94.0 | 131.5 |
| 81.2 | 101.7 | 127.1 |
| 100 | 106.7 | 124.2 |

[1] $T_m$ = crystal−nematic and [2] $T_i$ = nematic−isotropic transition temperatures in °C.

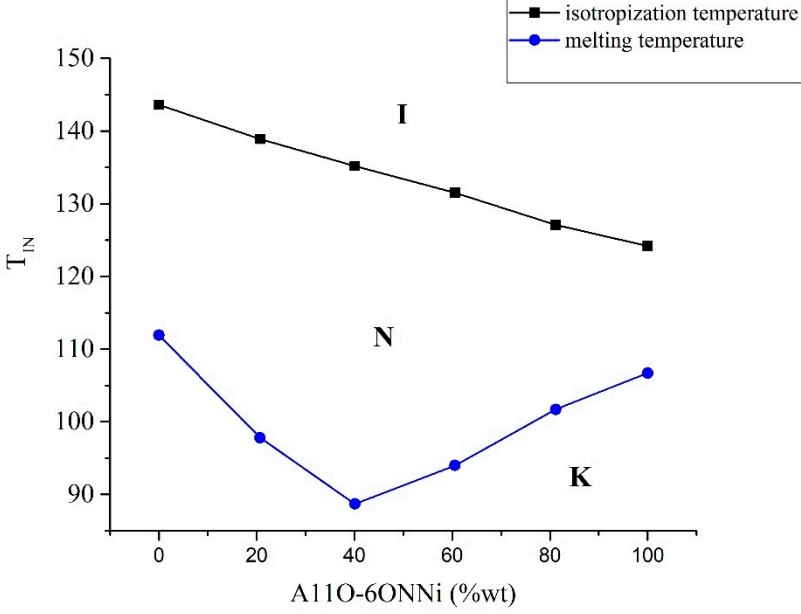

**Figure 7.** Binary phase diagram for A11O-6ONNi/A11O-6ONPd blends. K indicates the stability field of the crystalline solid phase, N that of the nematic phase and I the isotropic phase.

## 3. Materials and Methods

### 3.1. Materials

All starting products and solvents were commercially available. The compound 12-8NCu and the intermediate 0-8NCu were obtained as described in previous works [64].

### 3.2. Characterization

A Zeiss Axioscop polarizing microscope, implemented with an FP90 Mettler hot-stage, was used for optical observations. A Perkin Elmer Pyris 1 (DSC scanning calorimeter, PerkinElmer, Inc., Waltham, MA, USA) at a scanning rate of 10 °C/min, under nitrogen flow, was used to measure enthalpies and phase transition temperatures. All transition temperatures were measured in a second heating run. Decomposition temperatures (Td) were defined with a Perkin Elmer TGA 4000 (thermo-gravimetric analysis, PerkinElmer, Inc., Waltham, MA, USA), under nitrogen flow. The point at 5 wt.% weight loss is assumed to represent the decomposition temperature (Td). Copper content as CuO was obtained by thermos-gravimetric analysis at complete decomposition of metal complexes in oxygen atmosphere. A Bruker Avance II 400 MHz spectrometer was used to register $^{1}$H AND $^{13}$C NMR spectra. Mass spectrometry measurements were performed using a Q-TOF premier instrument (Waters, Milford, MA, USA) equipped by an electrospray ion source and a hybrid quadrupole-time of flight analyzer. Mass spectra were acquired in positive ion mode, in a 50% CH3CN solution, over the 200–800 m/z range. Instrument mass calibration was achieved by a separate injection of 1 mM NaI in 50% CH3CN. Data were processed by using MassLynx software (Waters, Milford, MA, USA). IR spectra were recorded using an FT-IR 4700LE spectrometer (JASCO, Tokyo, Japan) as a KBr pellet.

### 3.3. Synthesis of 12-8NCu

Synthesis of the ligand and of the copper complex are previously reported by us [57]. Here, we report the characterization data.

12-8N (ligand system)

1H NMR (400 MHz, Acetone-d6, 25 °C): δ = 8.55 (s, 1H), 8.12 (d, 2H), 7.46 (m, 1H), 7.11 (m, 3H), 6.79 (d,1H), 4.14 (m, 2H), 3.65 (m, 2H), 1.83 (m, 4H), 1.73 (m, 4H), 1.43–1.30 (m, 24H), 0.89 (t, 6H) ppm.

13C NMR (400 MHz, Acetone-d6, 25 °C): δ = 165, 164, 163, 155, 133, 122, 118, 116, 113, 111, 69, 59, 33, 32, 31, 28, 77, 23, 14 ppm.

Elemental analysis calculated (%) for C34H51NO4: C, 75.94; H, 9.56; N, 2.61; found: C, 75.66; H, 9.81; N, 2.43. MALDI-TOF *m/z*: 537.74 (M + H).

12-8NCu

Elemental analysis calculated (%) for C62H100N2O8Cu: C, 69.92; H, 9.46; N, 2.63; found: C, 69.84; H, 9.51; N, 2.08. MALDI-TOF of Ac1 *m/z*: 1065.8 (M + H). Copper content (calculated ad CuO% from TGA analysis): experimental = 7.62%; calculated = 7.47%.

### 3.4. Synthesis of A6O-8NCu and A11O-6ONPd

The synthetic procedure for these two compounds involves a two-step scheme (Scheme 2). The first, analogous to that used previously for the synthesis of 12-8NCu, provides for the formation of a bi-phenolic intermediate subsequently esterified with the appropriate acyl chloride. The synthesis of A6O-8NCu is reported below as an example.

**Scheme 2.** Reaction scheme of copper and nickel complexes.

First step: synthesis of complex bis [4-[(octylimino)methyl]-1,3-benzenediolato-copper (II) (**1**).

Complex (**1**) was synthesized by a stepwise addition of 1.00 g of n-octylamine dissolved in 10 mL of absolute ethanol to 1.07 (equimolar amount) of 2,4-dihydroxybenzaldehyde dissolved in 20 mL of absolute ethanol. The temperature was raised to boiling and then 0.5 g of sodium acetate and 0.700 g of copper (II) acetate monohydrate were added and the solution was kept boiling for other 5 min. The copper complex, that precipitate in a crystalline form on cooling, was recrystallized by ethanol. Yield 72%. Copper content as CuO: 14.2 per cent calculated; 14.5 per cent found.

Second step: Interfacial Synthesis of A6O-8NCu.

Under vigorous stirring, a solution of 1.260 g of 6-(4-(chlorocarbonyl)phenoxy)hexyl acrylate, obtained by literature method [64], dissolved in 50 mL of chloroform, was added to 0.745 g of 1 dissolved in a 100 mL water solution containing 0.300 g KOH and 2.0 g benzyltriethylammonium chloride. The mixture was left to react for about six minutes; then, 50 mL chloroform was added and the total chloroform phase was separated, washed with water, dried over sodium sulfate and concentrated to half of the volume by evaporation. To this solution, 130 mL hot ethanol was added. The final compound was separated as brown green crystals by cooled solution and purified by crystallization from ethanol/chloroform solution and successive column chromatography (Florisil, chloroform as eluent) and finally, crystallization. Yield 33%. Copper content as CuO: 7.2 per cent calculated; 7.3 per cent found.

A6O-8N (ligand system)

1H NMR (400 MHz, DMSO-d6, 25 °C): δ = 8.53 (s, 1H), 8.10 (d, 2H), 7.48 (m, 1H), 7.03 (m, 3H), 6.78 (d,1H), 6.43 (d,1H), 6.15 (m,1H), 5.78 (m,1H), 4.09 (t, 2H), 3.94 (t, 2H), 3.78 (t, 2H), 1.80–1.45 (m, 20H), 1.13 (t, 3H) ppm.

13C NMR (400 MHz, DMSO-d6, 25 °C): δ = 167, 166, 165, 163, 158, 155, 133, 131, 128, 121, 120, 114, 112, 69, 65, 62, 32, 30, 29, 27, 26, 23, 14 ppm.

Elemental analysis calculated (%) for C31H41NO6: C, 71.10; H, 7.89; N, 2.68; found: C, 70.95; H, 7.96; N, 2.49. MALDI-TOF *m/z*: 522.85 (M + H).

A6O-8NCu

Elemental analysis calculated (%) for C62H80N2O12Cu: C, 67.16; H, 7.27; N, 2.53; found: C, 65.91; H, 8.02; N, 2.44. MALDI-TOF of Ac1 *m/z*: 1108.9 (M + H). Copper content (calculated ad CuO% from TGA analysis): experimental = 7.25%; calculated = 7.17%.

### 3.5. Synthesis of A11O-6ONNi

The synthetic procedure for this compound was performed in boiling ethanol solution with a standard sequential procedure.

An amount of 0.500 g of 4-[4-(undecyloxy-11-acrylate)benzoyloxy]-2-hydroxybenzaldehyde was dissolved in 50 mL of hot ethanol. At this solution, the stoichiometric amount of 3-ethoxypropylamine was added in the presence of a large excess of sodium acetate. A stoichiometric amount of nickel perchlorate hexahydrate was then added to the reaction mixture, kept for a couple of minutes at its boiling point. The nickel complex precipitated as green crystals by cooled solution. The subsequent purification first involved crystallization from ethanol, followed by column chromatography (florisil, chloroform as eluent) and finally, crystallization with yield 30%.

To characterize the ligand system, a little amount of nickel complex (about 500 mg) was dissolved in 15 mL of chloroform. At this solution, 30 mL of water was added and some drops of HCl 37% was added under stirring. The initial green solution turns to light green. The chloroformic solution (uncolored) was separated from the aqueous solution, washed twice with water and dried with sodium sulphate anhydrous. The solvent was distilled and the white compound was crystallized by ethanol.

A11-6ON (ligand system)

1H NMR (400 MHz, DMSO-d6, 25 °C): δ = 8.53 (s, 1H), 8.11 (d, 2H), 7.46 (m, 1H), 7.05 (m, 3H), 6.79 (d, 1H), 6.42 (d, 1H), 6.13 (m, 1H), 5.78 (m, 1H), 4.09 (t, 2H), 3.98 (t, 2H), 3.73 (m, 4H), 1.80–1.25 (m, 22H), 1.13 (t, 3H) ppm.

13C NMR (400 MHz, DMSO-d6, 25 °C): δ = 167, 165, 164, 162, 157, 155, 133, 131, 130, 129, 122, 121, 115, 112, 70, 69, 67, 66, 58, 32, 30, 29, 26, 15 ppm.

Elemental analysis calculated (%) for C33H45NO7: C, 69.82; H, 7.99; N, 2.47; found: C, 70.34; H, 7.84; N, 2.51. MALDI-TOF m/z: 567.94 (M + H).

A11-6ONNi

1H NMR (400 MHz, DMSO-d6, 25 °C): δ = 8.15 (d, 4H), 7.49 (s, 2H), 7.34 (m, 2H), 7.11 (m,6H), 6.74 (d,2H), 6.41 (d,2H), 6.21 (m,2H), 5.81 (m,2H), 4.13 (m, 8H), 3.64 (m, 8H), 3.39 (m, 8H), 1.78–1.21 (m, 40H), 1.15 (t, 6H) ppm.

Elemental analysis calculated (%) for C66H88N2O14Ni: C, 66.50; H, 7.44; N, 2.35; found: C, 66.04; H, 7.15; N, 2.38. MALDI-TOF of Ac1 *m/z*: 1192 (M + H). Nickel content (calculated ad NiO% from TGA analysis): experimental = 6.26%; calculated = 6.41%.

A11-6ONPd

1H NMR (400 MHz, DMSO-d6, 25 °C): δ = 8.12 (d, 4H), 7.53 (s, 2H), 7.44 (m, 2H), 7.09 (m,6H), 6.68 (d,2H), 6.40 (d,2H), 6.15 (m,2H), 5.78 (m,2H), 4.11 (m, 8H), 3.61 (m, 8H), 3.44 (m, 8H), 1.75–1.25 (m, 40H), 1.11 (t, 6H) ppm.

4-[4-(undecyloxy-11-acrylate)benzoyloxy]-2-hydroxybenzaldehyde was obtained by direct esterification between the appropriate e carboxylic acids performed with standard procedures from alkyl chlorides [65] and 4-hydroxymethylbenzoate. This reaction produces two isomers (ortho and para substituted) that were previously separated by formation of Cu complex and subsequent decomplexation in a weak ethanolic acid solution.

## 4. Conclusions and Future Perspectives

The basic idea was to provide the mixing approach of some MOM model systems as a new strategy to qualify their use as commercial materials for application in electro-optical devices. Therefore, we investigated the miscibility of four MOMs (including three new ones) in binary systems. All the mixtures showed a perfect miscibility of the two

components throughout the composition range and the presence of a eutectic point with lower melting temperatures and a wider mesogenic range than the pure components.

Our future industrial R&D programs on the application of MOM as exceptional materials in display and photonic application will continue with the aim of further expanding the LC stability range, also improving the dielectric and electro-optics parameter. This goal could be achieved by dispersion of quantum dots (QDs) into mixtures of MOMs as host nematic liquid crystals. This strategy affects the regularity of the crystalline phase, causing lower melting temperatures and sometimes induces positive effects also on the isotropization temperature of the nematic phases [66].

**Supplementary Materials:** The following supporting information can be downloaded at: https://www.mdpi.com/article/10.3390/inorganics11010032/s1, Table S1. More relevant 1H NMR and IR data for ligand and related metal complexes; Figure S1: FTIR curves for of A6O-8N (black) and A6O-8NCu (red); Figure S2: FTIR curves for of A11O-6ON (black) and A11O-6ONCu (red); Figure S3: 1H NMR curves of A11O-6ONNi; Figure S4: 1H NMR curves for of 12-8N; Figure S5: 1H NMR curves for of A6O-8N; Figure S6: DSC thermograms of for A11O-6ONNi/A11O-6ONPd blends with composition of 0.207 wt./wt.: (black) first heating; (red) first cooling run and (blue) second heating run at 10 K/min scanning rate; Figure S7: DSC thermograms of for A11O-6ONNi/A11O-6ONPd blends with composition of 0.401 wt./wt.: (black) first heating; (red) first cooling run and (blue) second heating run at 10 K/min scanning rate; Figure S8: DSC thermograms of for A11O-6ONNi/A11O-6ONPd blends with composition of 0.606 wt./wt.: (black) first heating; (red) first cooling run and (blue) second heating run at 10 K/min scanning rate; Figure S9: DSC thermograms of for A11O-6ONNi/A11O-6ONPd blends with composition of 0812 wt./wt.: (black) first heating; (red) first cooling run and (blue) second heating run at 10 K/min scanning rate.

**Author Contributions:** Conceptualization, U.C.; Data curation, H.-A.H. and V.R.; Formal analysis, H.-A.H., V.R. and U.C.; Funding acquisition, U.C.; Investigation, V.R.; Methodology, H.-A.H., V.R. and U.C.; Project administration, U.C.; Resources, U.C.; Software, U.C.; Supervision, U.C.; Validation, V.R. and U.C.; Visualization, V.R.; Writing—original draft, H.-A.H. and U.C.; Writing—review & editing, U.C. All authors have read and agreed to the published version of the manuscript.

**Funding:** We gratefully acknowledge the financial aid provided by the Italian Ministry of Education, University and Research (MIUR), under grants PON PANDION 01_00375.

**Institutional Review Board Statement:** Not applicable.

**Informed Consent Statement:** Not applicable.

**Data Availability Statement:** Not applicable.

**Acknowledgments:** The authors would like to acknowledge the Electro-Optical Film Group of Snia Ricerche, Snia BPD (Fiat Group), Via Pomarico, Pisticci Scalo (MT), Italy.

**Conflicts of Interest:** The authors declare no conflict of interest.

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
