# Peer review of "Optical and Thermal Investigations of Eutectic Metallomesogen Mixtures Based on Salicylaldiaminates Metal Complexes with a Large Nematic Stability Range"

_inorganics, doi:10.3390/inorganics11010032_

Round 1
Reviewer 1 Report (New Reviewer)
In this work, the authors evaluated the mesomorphic behavior and the miscibility properties of binary mixtures of new series of Schiff base metallomesogen. To use as commercial materials for application in electro-optical devices, further expansion of LC temperature range and lowering transition temperature from N to K are required. The authors should describe the strategies the further expansion of LC temperature range and lowering transition temperature from N to K in revised version.
Author Response
Dear Editor,
We thank the Reviewers for the constructive comments on the manuscript.
As indicated below, we have checked all the general and specific comments provided by the Referees and have made necessary changes accordingly to their indications. We did our best to address all the questions as described in the following.
Reviewer
In this work, the authors evaluated the mesomorphic behavior and the miscibility properties of binary mixtures of new series of Schiff base metallomesogen. To use as commercial materials for application in electro-optical devices, further expansion of LC temperature range and lowering transition temperature from N to K are required. The authors should describe the strategies the further expansion of LC temperature range and lowering transition temperature from N to K in revised version.
According, we have added at lines 339-345, as future developments, a strategy to enlarge the mesofase stability range.
Reviewer 2 Report (Previous Reviewer 3)
The manuscript entitled "Optical and thermal investigations of eutectic metallomesogen mixtures based on salicylaldiaminates metal complexes with large nematic stability range" by Caruso and co-workers describe the mesomorphic behavior and the miscibility properties of binary mixtures of new series of Schiff base metallomesogen. The synthetic compounds were characterized by NMR, FT-IR and CHNX. Some reasonable conclusions were supported by sufficient experimental data. In my opinion, this work has decent scientific quality and nicely lies in scope of the Inorganics. The manuscript is acceptable after minor revision. Please consider the following some remarks which should be addressed by Authors:
1. In Results and discussion section (Page 3, line 89), "v(C-N)" should be replace by "v(C=N)". Schiff bases are constructed by azomethine (C=N) characteristic groups.
2. Use tables to display the IR and NMR results, for better understanding.
3. The corresponding HNMR spectra of Cu(II) and Ni(II) complex should be submitted as a separate supplementary material.
4. The title of Figure 2 (Page 4, line 95) is incorrect. Moreover, please you check the manuscript carefully.
Author Response
Dear Editor,
We thank the Reviewers for the constructive comments on the manuscript.
As indicated below, we have checked all the general and specific comments provided by the Referees and have made necessary changes accordingly to their indications. We did our best to address all the questions as described in the following.
Reviewer
The manuscript entitled "Optical and thermal investigations of eutectic metallomesogen mixtures based on salicylaldiaminates metal complexes with large nematic stability range" by Caruso and co-workers describe the mesomorphic behavior and the miscibility properties of binary mixtures of new series of Schiff base metallomesogen. The synthetic compounds were characterized by NMR, FT-IR and CHNX. Some reasonable conclusions were supported by sufficient experimental data. In my opinion, this work has decent scientific quality and nicely lies in scope of the Inorganics. The manuscript is acceptable after minor revision. Please consider the following some remarks which should be addressed by Authors:
- In Results and discussion section (Page 3, line 89), "v(C-N)" should be replace by "v(C=N)". Schiff bases are constructed by azomethine (C=N) characteristic groups.
According, we fixed the typo.
- Use tables to display the IR and NMR results, for better understanding.
According, we have added a table S1 in supplementary material.
- The corresponding HNMR spectra of Cu(II) and Ni(II) complex should be submitted as a separate supplementary material.
According, we have added figure S3 and 1H NMR chemical shift for compound at lines 315-317.
- The title of Figure 2 (Page 4, line 95) is incorrect. Moreover, please you check the manuscript carefully.
We thank the reviewer for this observation. We have fixed this and other typos in the text.
This manuscript is a resubmission of an earlier submission. The following is a list of the peer review reports and author responses from that submission.
Round 1
Reviewer 1 Report
Hakemi et al. described the preparation of several complexes (A6O-8NCu, A11O- 14 6ONNi and A11O-6ONPd) for electro-optical applications. This work could be published in this journal after solving the problems listed as following:
1- The title of manuscript did not describe the real work. I cannot find any electro-optical application through the manuscript.
2- I cannot follow and understand The first paragraph in the introduction section “In the last two decades the studies on the organic and organo transition metal com- 25 plexes due to the possibility of tuning their photochemical and photophysical properties 26 in the fields of organic optoelectronics and sensors for applications in several fields, for 27 instance photovoltaic devices [1-4], DNA binders [5-18], sensors [8,11-19], photosensitiz- 28 ers [19-21], molecular chemistry [22], fluorescent and colorimetric probe [23-37], medici- 29 nal chemistry [38] and metal-organic framework (MOF) construction [39-43]”. So, I strongly recommended the authors to rewriting it.
3- All figures need strong modify. All lines must be showed and information inside the figures must be written.
4- There are numerous mistakes in the English grammar. The authors are requested to improve the English grammar.
5- Several analyses are missing including FTIR, NMR, mass, and elemental analysis for the ligands and complexes.
6- If authors focused on the optical application of complexes, several analyses are required including UV, and fluorescence analyses.
Reviewer 3 Report
This submission by Hassan-Ali Hakemi and co-workers describes the synthesis and mesomorphically study of three MOM compounds (A6O-8NCu, A11O-6ONNi and A11O-6ONPd). The authors put the biggest emphasis on the mesophase stability range of three MOM compounds by DSC and TGA. However, the structure of Schiff base and three complexes have not been clearly characterized. Therefore, I recommend the publication of the paper after minor revisions given as follows: 1. FT-IR spectra of Schiff base and three complexes should be determined and IR spectra of the complexes must be interpreted in comparison with ligand data. 2. In Scheme 1, the azomethine nitrogen atoms (C=N) of Schiff base coordinated to metal ions should be confirmed by FT-IR or physical characterization. 3. I suggest the introduction should be concise.Author Response
Please see the attachment.

Reviewer 4 Report
This manuscript describes the preparation and DSC and TGA features of A6O-8NCu, A11O6NNi, A11O-6ONPd and their mixtures. Throughout the manuscript, the description is not easy to read, the sentences being poor grammatically. For examples,
Page1, lines 25-30: This part is not a sentence, being difficult to understand what the authors describe. This should be corrected.
Page 4, lines 119-120: What does “the main difference” mean? The authors should correct this sentence more explicitly.
As a whole, the authors should improve their description throughout the manuscript.
In the experimental section, the authors describe polarizing microscope and 1H NMR spectra. However, they did not describe the microscope and 1H NMR spectral features of their compounds. They should describe these features.
The authors claim their compounds are new. If so, analytical data of C, H, and N are necessary. They should describe these data also.
In the experimental section, significant figures (effective digits) of the amounts of reagents should be the same throughout the synthesis. Their description is not scientific, having many kinds of effective digits.
For example, Page 8, line 258, “0.5 g of sodium acetate” has only one effective digit, If so, the yield should be described as 7x10% not 72%.
There are many typos. For examples,
Page 8, lines 214-215: “complex bis[4-[(octylimino)methyl]-1,3-benzenediolato]copper(II)” not “Complex bis[4-[(octylimino)methyl]-1,3-benzenediolato-Copper (II)”
Page 8, line 219: “0.700 g of copper(II) acetate monohydrate” not“0.700g of Copper (II) acetate monohydrate”
Page 8, line 239: “0.500 g” not “0.500g”
Page 9, line 343, “The nickel complex precipitated” not “The Nickel complex precipitate”
The authors should correct many typographical errors throughout the manuscript.
Round 2
Reviewer 1 Report
The comments must be considered more seriously.
1- The title of the manuscript did not describe the real work. I cannot find any electro-optical application in the manuscript. I don't know why the authors add the potential word.
2- All figures need strong modify. All lines must be shown and information inside the figures must be written. The authors didn't change or modify.
3- There are numerous mistakes in English grammar. The authors are requested to improve the English grammar. The authors didn't modify it.
4- Several analyses are missing including FTIR, NMR, mass, and elemental analysis for the ligands and complexes. Such analyses are important to consider optoelectronic appliactions
Author Response
The comments must be considered more seriously.
1- The title of the manuscript did not describe the real work. I cannot find any electro-optical application in the manuscript. I don't know why the authors add the potential word.
In the article we describe the synthesis of new liquid-crystalline organometallic compounds and the study of their binary diagrams. The production of a device is out of the scope of this work, also if is well note that similar compounds are utilized for electro-optical application. For this reason, the authors believe that eliminating "Potential Materials for Electro-Optical Applications" would distort the spirit of the work.
2- All figures need strong modify. All lines must be shown and information inside the figures must be written. The authors didn't change or modify.
The figures have been modified in the first round. In the legends are reported the main information, inside the color line are associated to the compound. Other information inside would weigh down too much the graphic aspect of the figure.
3- There are numerous mistakes in English grammar. The authors are requested to improve the English grammar. The authors didn't modify it.
To remedy this problem, we will use the English editing service offered by the publisher.
4- Several analyses are missing including FTIR, NMR, mass, and elemental analysis for the ligands and complexes. Such analyses are important to consider optoelectronic applications.
All relevant analysis data characterizing new and already known compounds summarized in this work have already been reported in the first round of review (see page 10 lines 218-232, page 11 lines 264-275, page 12 line 294-309). However, to underline the close analogy of the FTIR results we added a further comment (line 88).
Reviewer 4 Report
The revised manuscript shows that the manuscript was revised according to the referees’ comments. But, please, check the following points and correct these.
Page 4 of 18, line 88: The description of “a red shift to 1902 cm-1” should be in error, because there no peak around this region in the IR spectra of the Cu complex in Figure 1. I guess that “1702 cm-1 not 1902 cm-1”.
Page 5 of 18, Figure 2: The scale and numerical figures are too small to read the peak positions. This should be revised for the readers.
Author Response
Thanks to reviewer comments.
The revised manuscript shows that the manuscript was revised according to the referees’ comments. But, please, check the following points and correct these.
Page 4 of 18, line 88: The description of “a red shift to 1902 cm-1” should be in error, because there no peak around this region in the IR spectra of the Cu complex in Figure 1. I guess that “1702 cm-1 not 1902 cm-1”.
The error was emended.
Page 5 of 18, Figure 2: The scale and numerical figures are too small to read the peak positions. This should be revised for the readers.
Accordingly the scale was modified.
Round 3
Reviewer 1 Report
As mentioned in the last two reports, the title of the manuscript did not describe the real work. "for electro-optical applications" means such materials are used for electro-optical applications. However, I did not see such applications in the manuscript. In addition, Figures are pretty poor, and fundamental analyses are not included. Finally, I can recommend this manuscript for publication.